# Effect of Growth Regulators on the Quality of Apple Tree Whorls

**Magdalena Kapłan [1], Kamila Klimek [2], Anna Borkowska [2,\*] and Kamil Buczyński [1]**

[1] Institute of Horticulture Production, University of Life Science, 28 Głęboka Street, 20-612 Lublin, Poland; magdalena.kaplan@up.lublin.pl (M.K.); kamil.buczynski@up.lublin.pl (K.B.)

[2] Department of Applied Mathematics and Computer Sciences, University of Life Science, 28 Głęboka Street, 20-612 Lublin, Poland; kamila.klimek@up.lublin.pl

[\*] Correspondence: anna.borkowska@up.lublin.pl

**Abstract:** High-quality nursery stock ensures the dynamic development of orchard crops. The most desirable by orchardists is a tree with a strongly developed crown, which makes it possible to shorten the investment period of newly planted orchards. The quality of the perianth and the degree of its branching depend on the growth strength of the rootstock, the ability of a given variety to form lateral shoots, the type and effectiveness of the treatments used to stimulate branching, soil and climatic conditions related to the location of the nursery and the course of the weather during the period of initiation and growth of young shoots. The study was conducted in 2017–2019 at a private nursery farm near Lublin. The aim of the experiment was to study the effect of mixtures based on the compounds benzyladenine (BA) and gibberellins ($GA_3$) and ($GA_{4+7}$) for chemical stimulation of lateral shoot outgrowth in apple cv. 'Alwa' and 'Najdared'. In the course of the research, significant effects of the type of growth regulations used and the variety on the height of the perianths, the number and length of lateral shoots and the degree of branching were demonstrated.

**Keywords:** apple tree; nursery stock; growth regulators; branch stimulation





## 1. Introduction

A prerequisite for the dynamic cultivation of orchard plants is high-quality nursery material. Its quality determines the earliness of entering the fruiting period, the yield and intensity of growth, and from the economic point of view, the profitability of the entire crop [1,2].

The basic parameter affecting the quality of a given nursery stock is the structure of the tree crown, i.e., the number of lateral shoots [1,2]. In addition, according to Makosz (2000) [3], it should be free of viruses and have a strongly expanded root system, so as to adequately supply the plant with water and nutrients it needs. The nursery stock most desired by orchardists is a tree with a strongly expanded crown, which makes it possible to shorten the investing period of newly planted orchards. Therefore, nursery men try to meet the expectations of orchardists by producing very good quality trees affecting the productive efficiency of the orchard in the following crop years [4–7]. However, the production of such a tree at the nursery stage still causes a lot of trouble for growers, since many popular apple tree varieties struggle to form crowns consisting of 5–8 lateral pedicels, and it is almost impossible to obtain it naturally. As a result, it is relatively difficult to obtain a very good quality oculant in a single season, so nursery men produce trees on a 3-year cycle (i.e., knip-boom) or on a 2-year cycle from grafting in hand [8–13].

The quality of a tree and the degree of its branching depend on internal factors, which include: the strength of the rootstock [5,9,10], the ability of a given variety to form lateral shoots [6] and external factors, i.e., the type and effectiveness of treatments used to stimulate branching, soil and climatic conditions associated with the location of the nursery and the course of the weather during the initiation and growth of young shoots [10–12].

The main reason for the lack of branching in the perianth is the strong apical dominance of the main shoot, which, depending on the variety, can greatly limit the growth of syleptic shoots, leading to the growth of trees with only the main shoot. Therefore, in order to prevent this phenomenon, treatments are used to improve the quality of nursery stock, the main purpose of which is to stimulate branching and obtain trees with a properly formed crown with a large number of lateral shoots. These include mecha-technique treatments, which involve pinching off the apical shoot and youngest leaves, twisting the top of the growth by 180°, and removing side shoots from the lower parts of the tree and rootstock [4,13]. These types of treatments are labor-intensive and require a large investment by the orchardist. Chemical treatments are also used using growth regulators, i.e., phytohormones, the effectiveness of which largely depends on the type of replacement and weather conditions during the application of the above-mentioned compounds [14]. The group of plant protection products called growth regulators includes a separate pool of products which, as the name suggests, are designed to correct and regulate plant growth. Phytohormones are a large group of organic compounds of natural or synthetic origin that modify the physiology of plants. This action involves stimulating or inhibiting specific physiological processes occurring in the plant, including development and growth processes, such as branching of maiden plants. The correct response of the plant does not depend on the concentration of a single phytohormone. It occurs as a result of quantitative changes between individual phytohormones [12]. In fruit nurseries, they are used for early defoliation of perianths in order to accelerate their digging and placing in cold stores and to stimulate the branching of shoots. Currently, the preparations available in this group in fruit nurseries can be applied to trees in the form of an aqueous solution (a much easier and less labor-intensive method) or lanolin paste for self-administration (a more precise method). This allows for precise intervention in the plant at a specific stage of its development, and as a result, it contributes to the achievement of better nursery material [15]. The purpose of the present study was to examine the effect of mixtures of growth regulators based on the compounds benzyladenine (BA) and gibberellins ($GA_3$) and ($GA_{4+7}$) for the chemical stimulation of lateral shoot outgrowth in apple trees of the 'Alwa' and 'Najdared' cultivars grafted on M.9 rootstock under the conditions of the Lublin Upland.

## 2. Materials and Methods

The research was carried out in 2017–2019 at a private nursery farm near Lublin. The experimental subjects were apple trees of 'Alwa' and 'Najdared' cv. budded on M.9 rootstock. The experiment was established in a randomized block design, including 3 combinations with 5 repetitions and 5 plants per plot annually. The experiment was established when the trees reached a height of about 70–75 cm.

The purpose of the experiment was to be able to test mixtures based on the compounds benzyladenine (BA) and gibberellins ($GA_3$) and ($GA_{4+7}$) for the chemical stimulation of lateral shoot outgrowth in apple perianths. Chemical treatments of branching stimulation, depending on the combination, consisted of application in the form of an aqueous solution of mixtures of preparations: Globaryll 100 SL (BA), Falgro Tablet ($GA_3$) and Regulex 10 SG ($GA_{4+7}$). The aqueous solution was sprayed on the 6 youngest well-developed lateral buds along with the leaves just below the growth cone. Treatments were carried out twice a season, the first application when the whorls reached an average height of about 70–75 cm, and the second treatment 10 days later. When using additional combinations: Control—trees not subjected to branching stimulation treatments;

BA (Globaryll 100 SL 6.8 mL) + $GA_3$ (Falgro Tablet 3.15 g/1 L water)—2 treatments;
BA (Globaryll 6.8 mL) + $GA_{4+7}$ (Regulex 10 SG 6.7 g/1 L water)—2 treatments.

During the course of the study, intensive tree protection against pests, diseases and weeds was carried out in accordance with the current plant protection program. During the growing season, tree height measurements were taken three times (the first on the day of the first application of growth regulators, the second 10 days later and thus on the day of

the second application of growth regulators, the third after another 10 days). In the autumn, before the trees were dug up, the diameter of the rootstock and whorl stems, height, and the number and length of lateral shoots were measured. Diameter measurements were made with a caliper, with an accuracy of 0.1 mm. The height of the perianths and the length of the lateral shoots were measured with a meter with an accuracy of 1.0 mm. The average shoot length and the sum of the shoot lengths were calculated on the basis of the experiments.

The results obtained in this test were statistically analyzed using a one-way analysis of variance and Tukey confidence intervals. The weather conditions results were presented graphically using a bar chart with standard deviation. The conclusions were based on the significance level of $p < 0.05$. Correlations between apple tree size and quality parameters were estimated by calculating Pearson's correlation coefficients. Techniques of multivariate analysis of experimental data were used, performing cluster analysis, in order to group objects into relatively homogeneous groups in such a way that objects in the same cluster were more similar to each other than to those in other clusters. The results of the cluster analysis were summarized using a dendrogram. All statistical analyses were performed using the SAS Enterprise Guide 5.1.

### 2.1. Characteristics of the Test Material

The M.9 rootstock is the most popular used for apple trees in Europe. It is a dwarfing rootstock fulfilling in modern intensive orchards and as a scarification insert. M.9 shoots are medium-long, straight, stout and reddish-yellow with gray or silver overgrowth. The leaves are large and lanceolate, shiny [16]. In the mother plant, up to 9 well-rooted rootstocks can be obtained from one plant. It is not resistant to low temperatures. The roots of M.9 are tender, and they are prone to detachment from the plant which attracts rodents. It is a rootstock with low susceptibility to stem base ring rot, while it is sensitive to powdery mildew and fire blight [7]. Varieties improved on this rootstock and enter fruiting early. Apples tend to ripen earlier compared to other rootstocks. It works well in the production of summer varieties. It is used by professional and amateur users and requires support due to its shallow root system. M.9 requires fertile, humus-rich and moist soils [4]. Many clones of this rootstock were created. The M.9 EMLA subclone had the ability to immunize against all viruses attacking M series rootstocks. T 337, T 339 and T 340 are subclones bred by the Dutch. French sub-clones are Pajam® Lancep and Pajam® Capiland, called Pajam 1 and Pajam 2. Due to the rapid development of nurseries and the emergence of large, modern nursery farms, research was conducted on more productive clones. Thus, these varieties were created: RN 29, F1 56, B1, B2 and B3 [7].

Apple variety 'Alwa'—a Polish variety bred at the Prussia Experimental Department in the 1950s. It was formed from free pollination of the 'Macoun' variety. The growth of the tree in the first four years after planting is very strong. The crown is loose and spreads. Lateral shoots are stiff, easily formed, with a poor number of short shoots [17]. Flower buds occur on short and long shoots. It blooms as one of the last cultivars. It is a very good pollinator for varieties flowering at a similar time. It fruits regularly and prolifically on dwarfing rootstocks [16]. Trees are tolerant to low temperatures, not very susceptible to powdery mildew and bark diseases, and moderately susceptible to scab. The harvest date is moderately early (mid-October). The storage term is movable depending on your storage facility after 3–9 months. 'Alwa' is a winter variety. The climate allows the variety to be grown throughout the country [17]. The variety has no tendency to form lateral shoots [6].

'Najdared' is a Polish apple variety bred as a mutant of the Idared variety in Zdzary. In the first years after planting, the tree grows strongly and has the ability to establish numerous short shoots on lateral branches [18]. The crown is thickened and spreads [19]. It fruits abundantly and annually. Fruits of medium size, under favorable conditions, are large, spherical, glossy skin with a distinct blush; the flesh is tastier than 'Idared' variety [20]. It is a winter variety. Depending on the season, fruits reach harvest maturity on October 10–20. The storage time is 3–9 months [21]. The trees are resistant to frost and

less resistant to powdery mildew and fire blight [22]. Pollinators of 'Najdared': 'Golden Delicious', 'Pinova', 'Ligol', 'Elstar', 'Shampion Reno 2' [6].

### 2.2. Characteristics of the Preparations

Globaryll 100 SL—an agent from the group of growth regulators, intended for chemical thinning of fruit buds, preventing the occurrence of the problem of alternate flowering of apple and pear trees and stimulating the propagation of seedlings in nurseries of fruit trees. The composition of the active substance content: 6-benzyladenine (the substance of the purine group)—100 g/L (9.50%).

Regulex 10 SG—an agent from the group of plant growth and development regulators in the form of water-soluble granules, designed to reduce fruit russeting, maintain fruit quality in apple trees and improve fruit settings in pear trees. Content of active substance: gibberellin $GA_{4+7}$ (plant growth regulator from the gibberellin group)—100 g/kg (10%).

Falgro Tablet—the regulator of plant growth and development in the form of water-soluble tablets for use in the cultivation of grapevine, rabarbar, ornamental plants and pear. Content of active substance: gibberellic acid $GA_3$ (a compound of the lactone group)—204 g/kg (20.4%).

## 3. Results

The study presents weather conditions, determining the average air temperature and total precipitation in the 2017–2019 growing season against the background of multi-year averages. It was found that the average air temperature in the study years was higher than the multi-year average. In 2018 and 2019, the average air temperature for the growing season was 2.5 °C higher than the multi-year average. A similar trend was shown for the analyzed weather parameter in all months of the growing season (Figure 1).

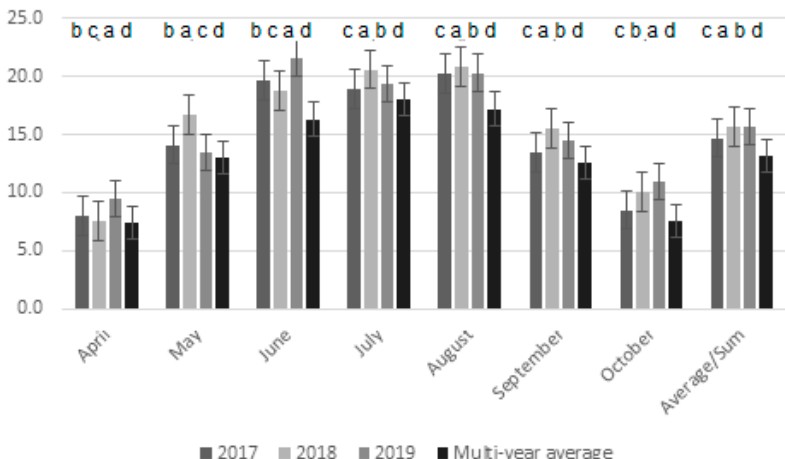

**Figure 1.** Average monthly air temperatures according to the agrometeorological station in Strzeszkowice, municipality of Niedzwica Duża in the months of April–October in the years 2017–2019 (Horti PROCAM). Significant difference a, b, c, d mean that different letters between columns indicate significant differences between survey years in each year at $\alpha = 0.05$.

The sum of precipitation in the analyzed study period was distributed differently depending on the growing season. In 2017, the sum of precipitation for the growing season was 292.0 mm higher than the multi-year average, in 2018 it was 61.0 m higher, while in 2019, it was equal to the multi-year average. The month with the highest precipitation in 2017 was June (282.0 mm), in 2018 July (124.0 mm) and in 2019 August (102.0 mm). The months with the least amount of precipitation were May (29.1 mm) in 2017, 2018 (36.0 mm) and 2019 (29.0 mm) October (Figure 2).

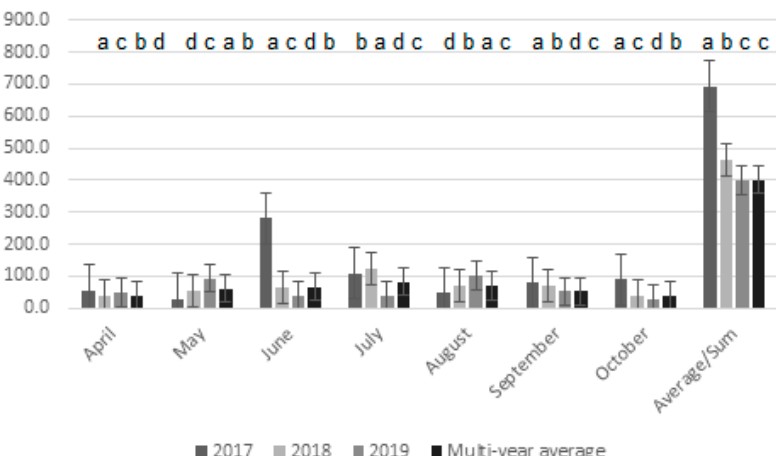

**Figure 2.** Rainfall totals according to the agrometeorological station in Strzeszkowice, municipality of Niedzwica Duża in the months of April–October in 2017–2019 (Horti PROCAM). Significant difference a, b, c, d mean that different letters between columns indicate significant differences between survey years in each year at α = 0.05.

The height of the perianths of two apple cultivars 'Alwa' and 'Najdared' was assessed at four dates, i.e., the first height measurement was carried out on the day of the first application of growth regulators; the second height measurement was carried out 10 days later and at the same time during the second application of growth regulators; the third height measurement was carried out 10 days after the second measurement and in autumn after the end of growth. Significant influence of the type of growth regulators applied and the variety on the studied trait was shown. The perianths of the 'Alwa' cultivar treated with BA + GA$_3$ were significantly ($p < 0.0001$) the highest in the first three height measurement dates, and during the autumn measurement, those treated with BA + GA$_{4+7}$. During the first and second height measurements, the perianths of the cultivar under study treated with BA + GA$_{4+7}$ were significantly ($p < 0.0001$) the lowest, while during the third and autumn measurements, the control perianths were the lowest. In the cultivar 'Najdared', during the first measurement in the control combination and in the subsequent measurement dates, the BA + GA$_3$ treated perianths were significantly ($p < 0.0001$) the highest of the tested combinations. In the case of this cultivar, the BA + GA$_{4+7}$-treated rootstocks in the first two measurement dates and the control ones during the third and autumn measurement dates were significantly ($p < 0.0001$) the lowest among the evaluated combinations (Table 1).

Another analysis was performed to check for significant differences between the tested varieties within each combination. In the case of control trees, regardless of the measurement date, the trees of the 'Najdared' variety were significantly taller than the Alwa trees ($p < 0.0001$). Trees of the 'Najdared' variety treated with growth regulators, regardless of the type of preparation, were significantly ($p < 0.0001$) taller than those of the 'Alwa' variety in the first three dates of height measurement; in autumn, this trend was reversed (Table 1). A significant effect of cultivar on the final height of apple trees was found in studies by Kaplan (2006) [15] and Kaplan and Baryla (2006) [23] regardless of the type of nursery stock. In these studies, both perianths and biennial trees of the cultivar 'Jonica' were taller than those of 'Shampion'. Klimek et al. (2018) [24] showed that growth regulators, the dose of their application, the type of rootstock and the year of the study can significantly modify the height of the perianths. The application of growth regulators can affect the transient growth behavior of the main shoot, but eventually, the trees gain sufficient height [23]. In a study by Kaplan et al. (2017) [25], transient growth retardation of saplings was shown in the 'Gloster' cultivar's whorls, while this effect was inconclusive in the 'Jonagold' cultivar. In the studies by Kapłan (2006) [15] and Kapłan and Baryła

(2006) [23] of most of the combinations analyzed, no transient effect of growth regulators on apple tree growth was observed in the period of two weeks after treatment.

Table 2 shows the parameters relating to the quality of the studied perianths, i.e., their bifurcation structure and thickness. The number of lateral shoots significantly ($p < 0.0001$) depended on the growth regulators used and the apple variety. Cultivar 'Alwa' treated with BA + $GA_3$ had the highest number of lateral shoots among the evaluated combinations, while the control had the lowest. The 'Najdared' cultivar treated with BA + $GA_3$ had significantly ($p < 0.0001$) better branching than the control. No significant differences were found between control trees and those treated with BA + $GA_{4+7}$. A significant effect of cultivar on the degree of branching of apple trees was shown. Najdared' apple trees, irrespective of the applied combination, formed a significantly ($p < 0.0001$) higher number of lateral shoots than 'Alwa'.

A study by Klimek et al. (2018) [23] showed that regardless of the year of the study, control trees produced significantly fewer lateral shoots than those treated with branching stimulating treatments. This is corroborated by the observations of Jaumien et al. (2000, 2004) [24,25] in which it was found that the whorls treated with growth regulators produced significantly more lateral shoots than the whorls of control trees. These authors also showed that Arbolin 036 SL (BA + $GA_3$) applied at a dose of 250 mg $\times$ $L^{-1}$ to 'Champion' cv. perianths had no significant effect on their branching [24]. A positive effect of growth regulators on the number of lateral shoots in numerous studies was shown by Jacyna (2002) [11] after two applications of a mixture of BA + $GA_{4+7}$ at a concentration of 250 ppm, Gąstol and Poniedziałek (2003) [26] spraying varieties: 'Alwa', 'Red Boskop' and 'Gloster', and Poniedziałek and Porębski (1995) [27], who showed a very clear and strong effect on the number of lateral shoots of BA + $GA_3$ and $GA_{4+7}$ and a slightly weaker but also significant effect of BA. Studies by Kaplan and Wociór (2005) [28] and Kaplan and Baryła (2006) [23] showed that variety has a significant effect on the degree of branching in apple trees. This phenomenon was observed in both control and growth regulator-treated trees. A study by Kaplan (2006) [15] showed that control okulants produced fewer lateral shoots than those treated with growth regulators, but this trend was not always significant and depended on the type of growth regulators and their concentration. The paper by Kaplan (2006) [15] subjected the above problem to an in-depth analysis based on the available literature and research over the past 30 years.

Statistical analysis for the evaluation of average lateral shoot length and the sum of lateral shoot length showed identical relationships. The maiden trees of 'Alwa' cv. treated with BA + $GA_{4+7}$ had significantly ($p < 0.0001$) longer shoots and the sum of shoot lengths among the evaluated combinations, while the control trees had significantly ($p < 0.0001$) the smallest. In the case of 'Najdared' trees treated with Ba + $GA_3$ the evaluated parameters were istotally the largest, while in the case of the control they were significantly ($p < 0.0001$) the smallest. Trees of 'Najdared', regardless of the combination used, formed significantly ($p < 0.0001$) longer shoots and a higher sum of lateral shoot lengths than 'Alwa' (Table 2).

In the research of Klimek et al. (2018) [24], it was shown that control maidens had longer syleptic shoots than after the application of growth regulators. These relationships were not confirmed in this paper. In a study by Jaumien et al. (2000, 2004) [26,27], it was shown that the total length of lateral shoots, as well as their number, is many times greater in okulants sprayed with growth regulators than in those not sprayed. Similar results were obtained by Gąstol and Poniedziałek (2003) [29]. In the study of Poniedziałek and Porębski (1995) [30], istotal effect on the sum of lateral shoot lengths was exerted by spraying with a mixture of BA + $GA_3$, while the application of BA had only a slight effect on increasing the sum of increments, since the resulting shoots are too short. In an earlier study by these authors [16], the growth regulators BA + $GA_{4+7}$ and BA + $GA_3$ affected the average sum of shoot lengths, but there were no significant differences between the applied preparations. This thesis was not confirmed in the present study. Slightly different results, which were fully confirmed in the present study, were achieved by Gudarovskaya and Szewczuk (2002) [10], who applied 'Arbolin' (Ba + $CA_3$) at a dose of 25 mL/1 L of water on

two-year-old trees of 'Alwa' and 'Gala' cultivars and achieved significantly higher sum of shoot length compared to the control. In a study by Kaplan and Baryła (2006) [23], it was shown that the average shoot length, as well as the sum of lateral shoot lengths, depended on the cultivar, but most often the effect was insignificant.

The perianths of the 'Alwa' cultivar treated with BA + $GA_{4+7}$ had significantly ($p < 0.0001$) the thickest rootstock and perianth stems, while those treated with BA + $GA_3$ had significantly ($p < 0.0001$) the thinnest. Trees of the 'Najdared' cultivar in the control and BA + $GA_3$ combinations did not differ significantly ($p < 0.0001$) in rootstock thickness, similar relationships were shown when evaluating the thickness of the trunks of trees treated with growth regulators. In most cases, a significant ($p < 0.0001$) effect of cultivar on the thickness of the evaluated apple trees was shown. Trees of 'Alwa' cultivar were significantly ($p < 0.0001$) thicker than those of 'Najdared', this relationship was not confirmed when evaluating the thickness of rootstock trunks in the BA + $GA_3$ combination (Table 2).

In the study by Klimek et al. (2018) [24], the shoots of control rootstocks did not differ completely from those treated with BA + $GA_{4+7}$, this was not confirmed in this study. The mentioned authors Klimek et al. (2018) [24] did not show significant differences between whorls treated with different mixtures of growth regulators; these relationships were not confirmed in the present study. In the study by Klimek et al. (2018) [24], it was found that trees of the 'Champagne Reno' variety treated with BA + $GA_{4+7}$ have significantly thicker trunks than in the control sample. This relationship was confirmed in this study only in the case of whorls of the 'Alwa' variety. Kaplan's (2006) [15] research did not show a significant effect of growth regulators on the diameter of the stems of the apple trees of the 'Shapion' and 'Jonica' cultivars.

Table 3 shows the height increments of apple tree whorls at 10 and 20 days and the total height increment after the first branching treatment. Trees treated with BA + $GA_{4+7}$ grew significantly ($p < 0.0001$) best among those evaluated. While the controls significantly ($p < 0.0001$) the weakest. Trees of cv. 'Alwa' regardless of the evaluation of the period of height growth. were characterized by significantly ($p < 0.0001$) higher growth than in Najdared. In trees treated with BA + $GA_3$ solutions, height increment in the first two terms showed. that 'Najdared' trees grew significantly better than 'Alwa' trees. This was not confirmed in the total height increment. The perianths of the 'Alwa' cultivar treated with BA + $GA_{4+7}$ in the two weeks after the branching treatment and throughout the growing season grew significantly ($p < 0.0001$) more than 'Najdared'. A different relationship was shown in the period of 20 days after the application of the 1st branching treatment (Table 3).

Table 4 shows the multivariate correlations of the two analyzed varieties. Below the value of 1 is presented the variety 'Alwa', while above 'Najdared' is presented. In the case of the cultivar 'Najdared', a strong positive correlation was shown between the last height of the whorls, and all three height increments and quality parameters of the whorls. An identical correlation was shown between all three measurements of height increment. and all analyzed parameters of the quality of whorls. It was shown. that as the diameter of the rootstocks increased, the diameter of the perianths and the average length of the lateral shoot increased. The number of lateral shoots was strongly correlated. with the average length and sum of lateral shoot lengths. Similarly, the average length of the lateral shoot with the sum of the lengths of the lateral shoots (Table 4a).

In the case of the cultivar 'Alwa', the final date of height measurement correlated significantly with all three height increment termini. The average length and total length of lateral shoots. The diameter of the perianths negatively affected the final height measurement and all height increments. The mean diameter of the rootstocks negatively affecting the ostseous height increment. The average length of lateral shoots significantly positively affected all three height increments and the number of shoots. while negatively affected the diameter of the whorls. The sum of the length of lateral shoots showed a similar relationship to the average length of lateral shoots, while the number of lateral shoots and the average length of lateral shoots were positively affected. (Table 4b).

**Table 1.** The influence of growth regulators on the height of apple trees of the Alwa and Najdared varieties during the growing season and in autumn after the end of growth.

| | Height of Maidens (cm) | | | | | | | | | | | |
| | I Application Growth Regulators | | | II Application Growth Regulators | | | 10 Days after II Application Growth Regulators | | | Final Height Measurement in Autumn | | |
| | 'Alwa' | 'Najdared' | *p*-Value | 'Alwa' | 'Najdared' | *p*-Value | 'Alwa' | 'Najdared' | *p*-Value | 'Alwa' | 'Najdared' | *p*-Value |
|---|---|---|---|---|---|---|---|---|---|---|---|---|
| Control | 73.6 Bb * | 85.6 Aa | <0.0001 | 91.7 Bb | 100.6 Ba | <0.0001 | 100.4 Cb | 110.6 Ca | <0.0001 | 134.2 Cb | 139.3 Ca | <0.0001 |
| BA + GA$_3$ ** | 76.2 Ab | 80.4 Ba | <0.0001 | 95.6 Ab | 101.8 Aa | <0.0001 | 105.3 Ab | 116.7 Aa | <0.0001 | 144.9 Ba | 144.7 Ab | <0.0001 |
| BA + GA$_{4+7}$ *** | 70.4 Cb | 74.0 Ca | <0.0001 | 91.2 Cb | 94.1 Ca | <0.0001 | 100.6 Bb | 110.7 Ba | <0.0001 | 151.1 Aa | 144.4 Bb | <0.0001 |
| *p*-value | <0.0001 | <0.0001 | | <0.0001 | <0.0001 | | <0.0001 | <0.0001 | | <0.0001 | <0.0001 | |

* Significant difference A, B, C—means that different letters in the column show significant differences between combinations within a given variety, rules a, b, c mean that different letters in a row show significant differences between varieties within a given combination with α = 0.05. ** BA (Globaryll) + GA$_3$ (Falgro Tablet). *** BA (Globaryll) + GA$_{4+7}$ (Regulex).

**Table 2.** Influence of growth regulators on the crown structure and thickness of apple cv. 'Alwa' and 'Najdared' pericarps.

| | Number of Shoots, szt. | | | Average Length of Lateral Shoot, cm | | | Total Length of Lateral Shoots, cm | | | Diameter of Trunks Shims—10 cm, mm | | | Trunk Diameter Whorls—30 cm, mm | | |
| | 'Alwa' | 'Najdared' | *p*-Value | 'Alwa' | 'Najdared' | *p*-Value | 'Alwa' | 'Najdared' | *p*-Value | 'Alwa' | 'Najdared' | *p*-Value | 'Alwa' | 'Najdared' | *p*-Value |
|---|---|---|---|---|---|---|---|---|---|---|---|---|---|---|---|
| Control | 0.5 Cb * | 8.3 Ba | <0.0001 | 8.9 Cb | 9.3 Ca | <0.0001 | 4.5 Cb | 79.2 Ca | <0.0001 | 22.6 Ba | 22.5 Ab | <0.0001 | 14.4 Ba | 13.6 Ab | <0.0001 |
| BA + GA$_3$ | 7.7 Ab | 13.4 Aa | <0.0001 | 12.9 Bb | 21.3 Aa | <0.0001 | 97.0 Bb | 288.7 Aa | <0.0001 | 22.5 Ca | 22.5 Aa | <0.0001 | 14.1 Ca | 13.1 Bb | <0.0001 |
| BA + GA$_{4+7}$ | 6.6 Bb | 8.3 Ba | <0.0001 | 16.6 Ab | 18.5 Ba | <0.0001 | 107.3 Ab | 154.7 Ba | <0.0001 | 23.8 Aa | 21.7 Bb | <0.0001 | 15.0 Aa | 13.1 Bb | <0.0001 |
| *p*-value | <0.0001 | <0.0001 | | <0.0001 | <0.0001 | | <0.0001 | <0.0001 | | <0.0001 | <0.0001 | | <0.0001 | <0.0001 | |

* Significant difference A, B, C—means that different letters in the column show significant differences between combinations within a given variety, rules a, b, c mean that different letters in a row show significant differences between varieties within a given combination with α = 0.05.

**Table 3.** Effect of growth regulators on the height increments of apple cv. 'Alwa' apple trees and 'Najdared'.

| | Height Gain in 10 Days after Application of I Branching Treatment | | | Height Gain in 20 Days after Application of I Branching Treatment | | | Total Height Gain after Application of I Branching Treatment | | |
| | 'Alwa' | 'Najdared' | *p*-Value | 'Alwa' | 'Najdared' | *p*-Value | 'Alwa' | 'Najdared' | *p*-Value |
|---|---|---|---|---|---|---|---|---|---|
| Control | 18.1 Ca * | 15.0 Cb | <0.0001 | 26.8 Ca | 25.0 Cb | <0.0001 | 60.6 Ca | 53.7 Cb | <0.0001 |
| BA + GA$_3$ | 19.4 Bb | 21.4 Ba | <0.0001 | 29.1 Bb | 34.3 Ba | <0.0001 | 68.7 Ba | 62.3 Bb | <0.0001 |
| BA + GA$_{4+7}$ | 20.8 Aa | 20.1 Ab | <0.0001 | 30.2 Ab | 36.7 Aa | <0.0001 | 80.7 Aa | 70.4 Ab | <0.0001 |
| *p*-value | <0.0001 | <0.0001 | | <0.0001 | <0.0001 | | <0.0001 | <0.0001 | |

* Significant difference A, B, C—means that different letters in the column show significant differences between combinations within a given variety, rules a, b, c mean that different letters in a row show significant differences between varieties within a given combination with α = 0.05.

**Table 4.** Multidimensional cooperation between all analyzed traits between each other with the division of 'Alwa' and 'Najdared' varieties.

| | Final Height Measurement in Autumn | I Measurement of Height Increment, cm | II Measurement of Height Increment, cm | III Measurement of Height Increment, cm | Diameter of the Rootstock, mm | Diameter of the Perianth, mm | Number of Shoots, pcs. | Average Length of Lateral Shoots, cm | Total Length of Lateral Shoots, cm | |
|---|---|---|---|---|---|---|---|---|---|---|
| Final height measurement in autumn | 1 | 0.9849 * | 0.9988 * | 0.9653 * | 0.7348 * | 0.5321 * | 0.8711 * | 0.9915 * | 0.9616 * | |
| I measurement of height increment, cm | 0.9844 * | 1 | 0.9753 * | 0.9959 * | 0.8411 * | 0.6706 * | 0.7729 * | 0.9994 * | 0.8987 * | |
| II measurement of height increment, cm | 0.9701 * | 0.9976 * | 1 | 0.9515 * | 0.7011 * | 0.4905 * | 0.8938 * | 0.9841 * | 0.9733 * | 'Najdared' (a) |
| III measurement of height increment, cm | 0.8495 * | 0.9289 * | 0.9526 * | 1 | 0.8864 * | 0.7347 * | 0.7127 * | 0.9913 * | 0.8556 * | |
| Diameter of the rootstock, mm | −0.4565 | −0.6056 | −0.6586 | −0.8572 * | 1 | 0.9652 * | 0.3071 | 0.8166 * | 0.5188 * | |
| Diameter of the perianth, mm | −0.9987 * | −0.9919 * | −0.9809 * | −0.8745 * | 0.5000 | 1 | 0.0478 | 0.6374 | 0.2775 | |
| Number of shoots, pcs. | 0.5421 | 0.3863 | 0.3223 | 0.0172 | 0.5000 | −0.5000 | 1 | 0.8007 * | 0.9728 * | |
| Average length of lateral shoots, cm | 0.9846 * | 0.9387 * | 0.9129 * | 0.7443 * | −0.2942 | −0.9748 * | 0.6805 * | 1 | 0.9171 * | |
| Total length of lateral shoots, cm | 0.8056 * | 0.6892 * | 0.6383 | 0.3713 | 0.1591 | −0.7754 * | 0.9345 * | 0.8967 * | 1 | |
| 'Alwa' (b) | | | | | | | | | | |

* Significant difference at $\alpha = 0.05$. I measurement height increment, cm 1—Height increment over a period of 10 days after application of the branching treatment. II measurement height increment, cm 2—Height increment in the period of 20 days after applying the branching treatment. III measurement Increase in height, cm 3—Increase in height after applying the branching treatment.

Table 5 shows the results of correlations between weather conditions. and the parameters of growth and quality of apple tree cankers, not depending on the variety studied. The analysis carried out did not show any significant correlations between the average air temperature and the sum of precipitation. and the traits determining the growth and quality of apple trees (Table 5). In the study of Klimek et al. (2018) [24], it was shown. that the weather conditions prevailing during the study had a significant effect on the degree of branching of apple tree cankers of the cultivar 'Shampion Reno'. Many authors [27,29,30] observed a clear effect of weather conditions on the action of chemicals and the formation of lateral shoots in apple tree perianths. which was not confirmed by the present study.

**Table 5.** Pearson correlation coefficient for weather conditions and parameters determining growth and quality of apple tree whorls.

|  | Temperature | Total Precipitation |
|---|---|---|
| I application of growth regulators | −0.3513 | −0.0874 |
| II application of growth regulators | −0.3063 | −0.1308 |
| 20 days after 1st application of growth regulators | −0.2175 | 0.0376 |
| Final height measurement | 0.1412 | −0.1158 |
| I measurement of height increment, cm | 0.3236 | −0.0363 |
| II measurement of height increment, cm | 0.1429 | 0.1684 |
| III measurement of height increment, cm | 0.3046 | −0.0176 |
| Diameter of the rootstock, mm | 0.0355 | −0.3881 |
| Diameter of whorls, mm | 0.1617 | −0.2198 |
| Number of shoots, pcs. | −0.1354 | −0.1077 |
| Average length of lateral shoots, cm | 0.0759 | −0.0445 |
| Total length of lateral shoots, cm | −0.0705 | −0.1286 |

Figure 3 shows a cluster analysis of the two varieties evaluated: 'Alwa' and 'Najdared'. The number of lateral shoots are analyzed, as well as the average length of lateral shoots and the sum of the lengths of lateral shoots. It can be observed that in both evaluated varieties the dendograms are arranged in two clusters each time. The first cluster consists of growth regulators used for branching, i.e., BA + GA$_3$ and BA + GA$_{4+7}$, while the single cluster is always the control. In this case, the unambiguous analysis of clusters confirms significant differences between the control and the formulations used. The application of any mixture of growth regulators. always significantly affected the branching parameters than no application of any product.

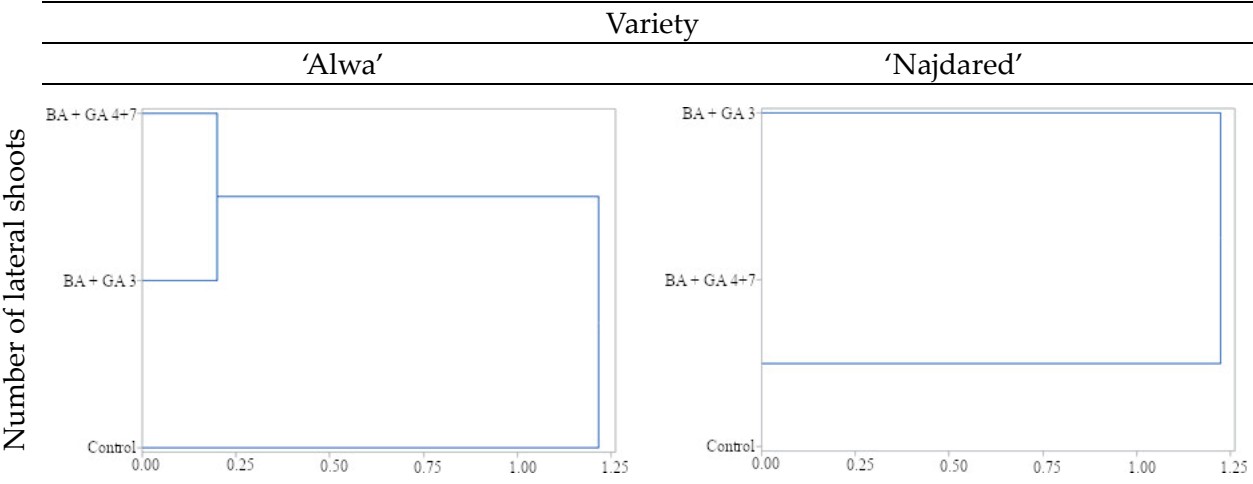

**Figure 3.** *Cont.*

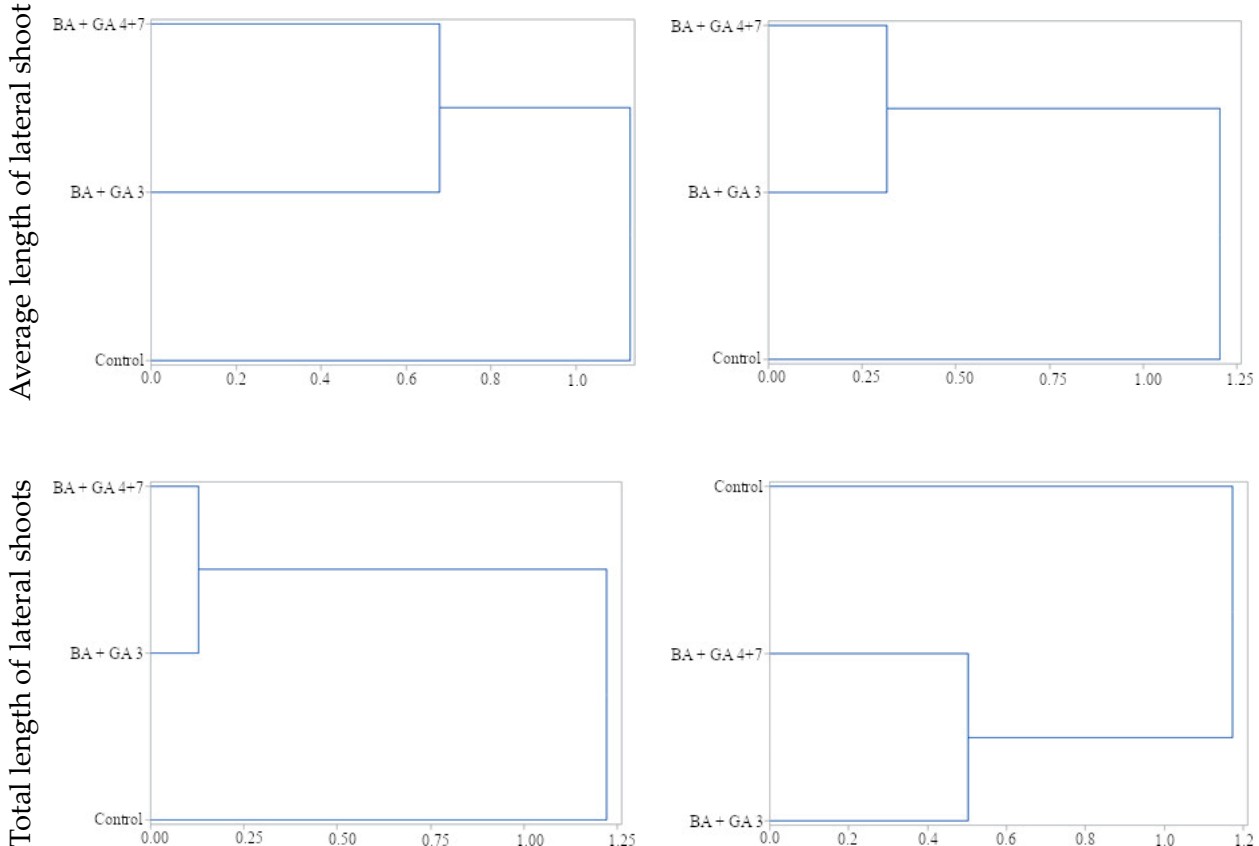

**Figure 3.** Principal component analysis of the crown structure of apple cvs'Alwa' and 'Najdared'.

### 4. Conclusions

Results indicate that the Alwa cultivar has been showing the best result of height in autumn and height increase with BA + $Ga_{4+7}$, after the first branching treatment and dia of trunk shim and trunk dia whorls. In contrast, the total length of lateral shoot, average length of lateral shoots and initial height gain with treatment were higher in Najdared cultivar.

1. The height of the studied perianths depended on the variety and the type of growth regulators used. The perianths of the cultivar 'Najdared' were independent of the combination in the summer were significantly higher than those of 'Alwa'. In the autumn, this trend was maintained only in the control combination. Trees treated with BA + $GA_3$ were significantly taller than after the use of BA + $GA_{4+7}$. The exception was the 'Alwa' perianth during the evaluation of growth in autumn.

2. The applied growth regulators had no significant effect on the diameter of the trunks of the rootstocks and apple trees of the 'Alwa' and 'Najdared' cultivars. This parameter significantly depended on the variety. The perianths of 'Najdared' were significantly thinner than those of 'Alwa' in most of the evaluated combinations.

3. The structure of the crown. which consists of the number. average length and the sum of the length of the lateral shoots of the studied perianths significantly depended on the variety and the branching treatment. The perianths of the cultivar 'Najdared' were characterized by significantly lower apical dominance and significantly higher susceptibility to the applied growth regulators than 'Alwa'. In most of the evaluated combinations, control trees had significantly less developed crowns than those treated with growth regulators. In the perianths of the cultivar 'Najdared', a clear significant effect of the type of growth regulators applied on the crown structure was shown. Trees treated with BA + $GA_3$ formed significantly better-developed crowns than after the application of BA + $GA_{4+7}$.

4. The height increments of the whorls are determined for the three periods. They significantly depended on the type of growth regulators applied. The perianths treated

with BA + GA$_3$ grew significantly weaker than after the application of BA + GA$_{4+7}$. A significant and unambiguous effect of cultivar on the studied parameter was shown in the case of control trees. where the perianths of 'Alwa' grew significantly more than those of 'Najdared'.

5. No significant correlations were shown between weather conditions. and traits determining the growth and quality of apple tree perianths.

**Author Contributions:** Conceptualization, M.K. and K.K.; methodology, M.K. and K.K.; software, K.K.; validation, A.B. and K.B.; formal analysis, M.K. and K.K.; investigation, A.B. and K.B.; resources, M.K.; data selection, M.K. and K.K.; writing—preparation of an original project, M.K. and K.K.; writing—review and editing, A.B. and K.B.; supervision, M.K.; obtaining financing, M.K. and K.K. All authors have read and agreed to the published version of the manuscript.

**Funding:** Statutory activity of the University of Life Sciences in Lublin.

**Institutional Review Board Statement:** Not applicable.

**Informed Consent Statement:** Not applicable.

**Data Availability Statement:** Not applicable.

**Conflicts of Interest:** The authors declare no conflict of interest.

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
