# Peer review of "Effect of Growth Regulators on the Quality of Apple Tree Whorls"

_applsci, doi:10.3390/app132011472_

Round 1
Reviewer 1 Report
In this research paper some improvement is needed suggestions are mentioned in original paper and below this. After this improvement paper may be published.
Introduction:
Line 36 - nursery men
Line 50-51 - something are missing
Line 57-58 - Incomplete the sentence
Line 59-72 – Rewrite the paragraph
Materials and Methods
Line 78-80 – Rewrite the sentence in meaningful.
· Reduce the Characteristics of the test material
· Write the methods how you conduct the experiment.
· What observation was recorded and methods of observation
Results:
Line 173-174 - April - October, 2017 or 2019.
· Improvement is required
Discussion:
· Discuss all observation how response and support with current reverence.

Some part of text are required english improvement
Author Response
Thank you very much for reviewing:
Ad. 1 Corrected as suggested
Ad. 2 Corrected as suggested
"The main reason for the lack of branching in the perianth is the strong apical dominance of the main shoot, which, depending on the variety, can greatly limit the growth of syleptic shoots, leading to the growth of trees with only the main shoot" completed in lines 50-52
Ad. 3 Corrected as suggested, supplemented in lines 60-63
Ad. 4 Corrected as suggested, supplemented in lines 65-79
Ad. 5 Corrected as suggested, cauliflowered changed to Budded line 99
Reviewer 2 Report
The manuscript is interesting, but it should improve. The references should actualize. The more actual is for 2018.
In the section of methodology, it is very important add the statistical analysis. ¿What experimental design used? ¿What technique to comparison mean used.
The figures 1 and 2 the authors must put the letters to shower the significant differences (P<0.05)
In the results are important to add after the words significant (p<0.05) or (p>0.059) according each case. Example. Line 204 Another analysis was carried out to check for significant differences (P<0.05) between
Author Response
Thank you very much for reviewing:
- The manuscript is interesting, but it should improve. The references should actualize. The more actual is for 2018.
The presented research covers a 3-year research cycle. In order to reliably present the impact of the applied treatments on the quality of apple trees at the nursery stage, the average from three years of research was taken into account. The study presents averages from three years of research because at the stage of statistical analysis no significant differences between the studied periods were demonstrated. Therefore, these results were used as replicates in further statistical analysis.
- In the section of methodology, it is very important add the statistical analysis. ¿What experimental design used? ¿What technique to compare mean used.
The description of the statistical analysis has been completed
- The figures 1 and 2 the authors must put the letters to shower the significant differences (P<0.05)
The letters in Figures 1 and 2 have been supplemented by analyzing the differences in individual years of research and the multi-year average for each month separately.
- In the results are important to add after the words significant (p<0.05) or (p>0.059) according to each case. Example. Line 204 Another analysis was carried out to check for significant differences (P<0.05) between
Thank you for your attention, this has been taken into account in the descriptions of each table
Reviewer 3 Report
Dear editor,
Thanks for allowing me to review the paper --------- “Effect of growth regulators on the quality of apple tree whorls. Authors were making great efforts to see the effect of different growth regulator combinations to see the effect on apple seedling growth in terms of height, perianth and whorls growth. However, the authors did a lot of effort to see the effect of growth hormones on the growth of two indigenous cultivars Alwa and Najdared.
The concept of the work is good, but the execution and way of writing are very poor. Pointwise comments are given below. Comments English writing is extremely poor.
1. In line 91, the reviewer doesn’t understand the language other than English written in the text.
2. Line 110, significance is based on p<0.005. Is it true? If it is true, I would like to see the statistical analysis done with the data in the tables.
3. Although weather condition directly affects the growth of the plants, according to the authors here, in this case, whether condition has not correlated with treatments, so what is the meaning of these two rainfall and air temperature data here, it seems that they do not affect the growth of the seedlings here, Fig. 1and Fig.2 should go in the supplementary and result with this section should be reduced.
4. Lines 194-202, please elaborate on it more accurately and precisely. This is the main result of the experiments, and it has explained very badly
5. Line 204-205 Significant difference of tested varieties, in which term ??, again the explanation is miswritten pls improve it.
6. Line 259, again in a different language????
7. Line 286-295 similar problem, different language???
8. 298-299, and the barley after the end of growth?????, also table 1 different language??
9. Results indicate that the Alwa cultivar has been showing the best result of height in autumn and height increase with BA+Ga4+7, after 1st branching treatment and dia of trunk shim and trunk dia whorls. In contrast, the total length of lateral shoot and average length of lateral shoots and initial height gain with treatment were higher in Najdared cultivar. This would be total conclusion of both the cultivars with their possible explanation. The author should have explained the data precisely.
10. Overall, I would suggest the authors should have go writing part with some native English speaker and do the explanation of the dat very precisely.
11. The author should provide the statistical analysis report they have done using SAS analysis software.
-
Author Response
Thank you very much for reviewing:
- In line 91, the reviewer doesn’t understand the language other than English written in the text.
Corrected as per reviewer's suggestion
- Line 110, significance is based on p<0.005. Is it true? If it is true, I would like to see the statistical analysis done with the data in the tables.
Corrected as per reviewer's suggestion
- Although weather condition directly affects the growth of the plants, according to the authors here, in this case, whether condition has not correlated with treatments, so what is the meaning of these two rainfall and air temperature data here, it seems that they do not affect the growth of the seedlings here, Fig. 1and Fig.2 should go in the supplementary and result with this section should be reduced.
We agree with the reviewer's suggestion that weather conditions during the application of growth regulators have a significant impact on the effectiveness of these treatments. The weather conditions shown in Fig. 1 and 2 apply to the entire growing season because the average air temperature and rainfall throughout the entire growing season may modify the growth and quality parameters of apple trees. Additionally, a statistical analysis of both assessed weather parameters was performed for each month, separately analyzing the differences between years.
- Lines 194-202, please elaborate on it more accurately and precisely. This is the main result of the experiments, and it has explained very badly
Corrected as per reviewer's suggestion
- Line 204-205 Significant difference of tested varieties, in which term ??, again the explanation is miswritten pls improve it.
Corrected as per reviewer's suggestion
- Line 259, again in a different language????
Corrected as per reviewer's suggestion
- Line 286-295 similar problem, different language??
Corrected as per reviewer's suggestion
- 298-299, and the barley after the end of growth?????, also table 1 different language??
Corrected as per reviewer's suggestion
- Results indicate that the Alwa cultivar has been showing the best result of height in autumn and height increase with BA+Ga4+7, after 1st branching treatment and dia of trunk shim and trunk dia whorls. In contrast, the total length of lateral shoot and average length of lateral shoots and initial height gain with treatment were higher in Najdared cultivar. This would be total conclusion of both the cultivars with their possible explanation. The author should have explained the data precisely.
Thank you very much for the excellent summary of the most important results of your work, which is included in the summary section
- Overall, I would suggest the authors should have go writing part with some native English speaker and do the explanation of the dat very precisely.
Thank you very much for your insightful review. Language errors have been corrected
- The author should provide the statistical analysis report they have done using SAS analysis software.
The first 3 tables contain one-factor analyzes considering differences between varieties and combinations. I attach the results of these analyzes in a separate file
Round 2
Reviewer 2 Report
The corrections were done
Author Response
Good morning
Thank you very much for your review
Reviewer 3 Report
N/A
Author Response

(The authors gave the same response as above.)
